# The Activation Energy of Strain Bursts during Nanoindentation Creep on Polyethylene

**DOI:** 10.3390/ma16010143

**Published:** 2022-12-23

**Authors:** Mohammad Zare Ghomsheh, Golta Khatibi

**Affiliations:** Christian Doppler Laboratory for Lifetime and Reliability of Interfaces in Complex Multi-Material Electronics, CTA, TU Wien, Getreidemarkt 9/164, 1060 Vienna, Austria

**Keywords:** activation energy, activation volume, nanoindentation, creep, strain burst, polyethylene

## Abstract

In the present investigation, statistical characterization of strain bursts observed during the load-controlled deformation of high-density polyethylene (HDPE), which arise within the crystalline phase during plastic deformation, was carried out via high-resolution nanoindentation creep experiments. Discrete deformation processes occurred during the nanoindentation creep tests, which indicated that they arose from the break-off of dislocation avalanches, i.e., dislocation climb is a possible mechanism for indentation creep deformation. Characterization of the strain bursts, in terms of the associated height and number, demonstrated that these quantities followed a Gaussian distribution depending on the load and loading rate. This analysis enabled the accurate measurement of creep activation energy. Our method used nanoindentation tests to measure the creep activation energy of HDPE within both the crystalline and amorphous phases. The activation energy of the creep process within the crystalline phase was evaluated using two methods. The frequency of jumps within the crystalline phase, as a function of the strain rate, showed two peaks related to the 5 nm and 10 nm jump sizes that corresponded to the block size within the crystalline lamellae. The results indicated that the intervals coincided with the mean free path of dislocations and the block grain boundaries acted as dislocation barriers. From the dependence of burst frequency on the strain rate and temperature, the activation energy and thermally activated length of the dislocation segment for the plastic slip activation were determined to be 0.66 eV and 20 nm, respectively. Both numbers fit well to the Peterson’s model for the nucleation and motion of thermally activated dislocation segments. A similar activation energy resulted from the differential mechanical analysis of the literature for the α_I_—transition, which occurred near room temperature in polyethylene. The transition was described as the generation of screw dislocation and its motion along a block grain boundary; therefore, this process is suggested to be the basic mechanism underlying the strain bursts observed in this study.

## 1. Introduction

The identification of deformation mechanisms in polymers is performed using mechanical tests, which characterize the process macroscopically and allow microstructural observations of the deformed samples. This enables obtaining information about the deformation processes, especially plastic deformation, that facilitates the identification of deformation mechanisms. Under the influence of external stress, plastic deformation occurs in many crystalline materials due to dislocation motion. On a microscopic scale (i.e., in the plastic deformation of microcrystals), dislocation avalanches lead to jumps in creep curves or stress-strain curves, which are known as deformation jumps or strain bursts, whereas plasticity appears as a smooth curve on a macroscopic scale. On the other hand, the probability of such events occurring increases with the crystallinity of the polymer [1], i.e., they depend on the degree of crystallization, and no strain bursts have been observed in amorphous material [2,3]. Therefore, strain bursts are likely to arise within the crystalline phase.

Nucleation, generation, and motion of dislocations in the plastic deformation of crystalline materials are important phenomena that have been highly investigated in metals, but few investigations have been conducted on semicrystalline polymers. X-ray line profile analyses [4,5,6] have provided reliable proof of the participation of dislocations in the crystalline phase during the plastic deformation of semicrystalline polymers [7,8,9,10,11,12], as well as confirming that they play a main role at the nanometer and molecular scale. Furthermore, the number of methods capable of performing such investigations is quite small [1]. It should be noted that other mechanisms, such as adiabatic melting and recrystallization, were suggested by Flory and Yoon [13] to play minor roles in quasi-static conditions. Determining the mechanical properties to better understand plastic deformation is more complex for polymers than for many other materials since polymers exhibit a pronounced time dependence in their response to load. The microstructure of polymers consists of long chains that make up three typical areas: (i) the crystalline lamella, (ii) the interface, and (iii) the amorphous region. The amorphous phase in polyethylene at room temperature can be considered as rubbery or quasi-liquid. Interaction between the amorphous and crystalline phases, orientation, and confinement lead to a stiffening of this rubbery phase, and recent literature states an apparent modulus of approximately 300 MPa [14]. The strength, as such, is mostly controlled by the crystalline phase since the viscosity of the amorphous phase can be neglected in the rubbery and quasi-static deformation regime. Tie molecules located in both phases act as stress transmitters between the crystalline lamellae. Because of this, deformation mechanisms in polymers are rather complex with several mechanisms occurring simultaneously with respect to the macroscopic deformation. However, understanding and identifying the molecular processes that control the elastic properties, strength, and ductility of polymers are necessary to improve the predictive abilities of models and finally, the quality of materials [15].

There is a wide range of opportunities for the use of the nanoindentation technique to measure the mechanical properties of materials. One of the important applications of nanoindentation is measuring the mechanical properties of polymers. It is necessary to establish a reliable analysis method to characterize polymers owing to their low stiffness, remarkable creep during loading and holding, and extensive recovery upon unloading. Indirect evidence for the occurrence of such bursts during plastic deformation has been provided by systematic high-resolution nanoindentation creep studies with low load, e.g., quantitative analyses of stress and/or strain bursts [16]. Typically, during nanoindentation tests, forces in the mN range and penetration depths in the nm range are applied. Thus, this method has the potential to study discrete atomic rearrangements (i.e., strain bursts/jerky motion mechanism) under stress, especially during plastic deformation [17,18,19,20,21], thus improving the understanding of the molecular and atomistic mechanisms related to dislocation formation and shedding some light on dislocation-mediated plastic deformation. In metals, strain bursts observed during nanoindentation are associated with dislocation events [22,23]. Polymer crystals are also known to deform by dislocations. Therefore, this study takes advantage of the nanoindentation technique to quantify and characterize strain bursts during the creep deformation of HDPE. Investigation of the dislocation effects on the initiation of plastic deformation, which is called plastic creep, was performed using nanoindentation testing. 

The occurrence of dislocation avalanches from strain bursts during nanoindentation creep testing of HDPE has been studied. Such strain bursts are well known in the plastic creep deformation of non-polymeric materials and have been repeatedly identified as an avalanche-type movement of dislocations [17,18,20,21]. In this work, we continued our previous work [24] and high-resolution nanoindentation creep studies were performed on as-received HDPE samples in order to study the dislocation kinetics of HDPE. In the last work, polyethylene samples were subjected to different degrees of plastic deformation during nanoindentation and equivalent indentation strains at shallow depth were calculated using contact mechanical analysis of the indentation process. These results indicated that the dislocation density increased with increasing plastic deformation and the extent of strain bursts was found to be related to the thickness of the lamellae. The number and height of bursts were quantitatively evaluated as a function of the loading rate and load applied in the nanoindentation creep tests, and the results were clearly interpreted in terms of dislocation movement [24]. These strain bursts correspond to the activation of atoms that are stuck behind barriers. Here, the activation energy and activation volume of the creep process within the crystalline phase were evaluated using the Arrhenius equation via two methods. Furthermore, the activation energy of the creep process within the amorphous phase was evaluated using the Arrhenius equation. The results were compared with the those of other studies, as well as the recent investigation conducted by Wilhelm [25], who carried out high-resolution torsion creep tests using a rheometer on bulk samples of HDPE, reporting very similar strain bursts to the strain bursts observed by nanoindentation creep tests.

## 2. Material and Experiments

Polyethylene has a relatively simple crystallographic structure; therefore, it is an ideal model material to study the mechanical properties of semicrystalline polymers. Additionally, it also allows varying the lamellae thickness under selected crystallization times and temperatures over a wide range by crystallizing under elevated pressure [11,26]. Hence, polyethylene was investigated in the present work. To reduce the influence of texture, a pressure plate of a high-density polyethylene copolymer (BorSafe HE3492-LS from BOREALIS) with a density of 952 kg/m³ and thickness of 10 mm was produced. The lamella thickness and crystallinity values were determined by small angle X-ray scattering (SAXS) and differential scanning calorimetry (DSC). DSC can quickly and easily determine the crystallinity of polymers by measuring the enthalpy (heat of fusion) of polymers and the precision of DSC is typically a few percent. The thermal analysis of the samples was performed using a calorimeter Perkin Elmer Pyris DSC-8500 and the percentage of crystallinity was found to be 45.3 ± 0.2% [24]. The Gibbs-Thomson [27] equation was used to determine the lamellae thickness using the DSC method, and the obtained value was 19.4 ± 0.3 nm. SAXS was also used to verify that value. Diffraction patterns were recorded on a Bruker AXS Nanostar equipped with a Vantec 2000 area detector. By evaluation of the radial distribution function, a lamellar thickness of 20.4 ± 0.1 nm was determined [28]. The differences between the crystallinity values obtained using the two methods are believed to be due to the different nature of the two methods. The crystallinity obtained by DSC is based on the enthalpy of the fusion of polymer crystals, whereas the crystallinity obtained by SAXS is based on scattering intensity peaks. Therefore, these two techniques reflect aspects of the crystalline phase that are fundamentally different from each other. Thus, differences between the crystallinity values obtained by DSC and SAXS can be expected. For the nanoindentation tests, disc-shaped samples were prepared with a thickness of 0.8 mm and radius of 8 mm, and then mechanically polished with P4000 grit sandpaper (premium SiC abrasive) for at least 10 min with water cooling and low feed rate, and then the samples were cleaned using ethanol to reduce surface effects [24]. Surface roughness (Ra) was less than 20 nm and therefore did not influence the nanoindentation measurements. The indentation scheme shown in Figure 1 consisted of four steps: (1) a loading step to the maximum load, (2) a holding step under the maximum load for a while, (3) an unloading step to a very small load (5% of maximum load), and (4) a final holding step for a while to record the thermal drift rate. All of these curves were recorded for data analysis [29]. The high-resolution nanoindentation creep experiments were performed in a load-controlled manner (while the indenter is pressed into the sample by continuously and simultaneously measuring the load and the depth), as shown in Figure 1, using an ASMEC-Universal Nanomechanical Tester UNAT. The displacement and force noise level of the nanoindenter were below 0.5 nm and 3 μN, respectively, while the digital displacement and force resolution were 0.002 nm and 20 nN. Measurements were performed using a three-sided pyramidal Berkovich indenter, and calibration was conducted in terms of indenter stiffness and contact area using fused silica and sapphire standards, according to the methods described by Oliver and Pharr [30] and Balint [31].

## 3. Results

To verify if the creep experiments were carried out in the plastic deformation regime, a comparison of the experimental loading curves with the Hertzian elastic contact theory [32] was carried out. At a very small depth, the shape of the Berkovich tip appeared to be rounded. Assuming purely elastic tests, the load (*P*) is related to the indentation depth (*h*) during elastic loading and can be calculated according to the Hertz equation [32] using the following expression:(1)$$P=\frac{4}{3}E_{r}R^{1/2}h^{3/2}$$
where *P* is the load, *R* is the radius of the indenter tip (967 nm), and *E_r_* is the reduced modulus, for the specimen and indenter materials. With respect to the obtained values of the nanoindentation creep tests (load, indentation depth, and reduced modulus value determined using the nanoindentation machine), the elastic response according to the Hertzian elastic contact theory was determined by Equation (1).

Equation (1) was plotted in Figure 2, with the experimental nanoindentation curve represented by the blue line. As seen in this figure, the initial elastic portion of the load-displacement curve fit well with the Hertzian elastic contact theory (red line). The good fit indicated the elastic regime in the loading curve i.e., it showed the portion of the purely elastic response of the indentation before the occurrence of plasticity. A comparison of this graph with that of the experimental indentation (blue line) showed that all experiments occurred close to the plastic regime. In other words, the separation of the plastic and elastic contributions to deformation revealed that at a critical load and loading rate, plastic deformation occurred in the creep segment, i.e., strain bursts occurred in the plastic deformation segment. Fragments of the creep rate curve in terms of time for the two PE samples studied are shown in Figure 3. In the creep segment, strain bursts were monitored and statistically evaluated [24]. In these figures, detailed recorded creep curves were shown for a time segment of about 500 s. In all obtained creep rate curves, besides the displacement jumps, noise type fluctuations in creep rate were also observed (Figure 3). Careful inspection of the curves obtained showed that the fluctuations were below 1 nm/s (this threshold is shown in figures as a dashed line), which was a threshold value depending on the temperature and load. Since the slope of the creep curve was usually high at the beginning of the creep section, no jumps were considered in this regime and the first 20% of creep time was neglected. In order to separate displacement jumps from noise fluctuations, only those jumps were considered as strain bursts which were significantly larger than 1 nm/s. In this typical plot (Figure 3b), negative strain bursts (red arrow) were probably caused by restoration forces from the amorphous phase. Therefore, they were not considered in our investigations. Thus, by these experiments, the total number of jumps exceeding a certain threshold was obtained.

Figure 4 shows the frequency distribution of jumps within both the crystalline and the amorphous phases, as a function of the strain rate. Several peaks were observed: the first peak (biggest) showed the frequency of jumps within the amorphous phase related to the 1 nm jump sizes, while the other peaks showed the frequencies of jumps within the crystalline phase related to the 5 nm and 10 nm jump sizes.

### 3.1. Activation Energy

Now the question is, which mechanism or mechanisms are operating in the stress-induced generation of dislocations in semicrystalline polymers? The activation energy and activation volume gave us interesting insight into this topic. In this work, the activation energies connected with the strain bursts observed in the nanoindentation creep curves of the crystalline phase of HDPE were calculated using different methods. The flow creep rate in solids ($\dot{\varepsilon }$) to describe thermally activated plastic flow depends on the temperature (*T*) and mechanical stress (*σ*), as given by (Arrhenius equation) [33,34];
(2)$${\left.\dot{\varepsilon }={\dot{\varepsilon }}_{0t}\exp\left(\frac{-Q_{t}}{k_{B}T}\right)\right|}_{\sigma =const.}$$
where *k_B_* is the Boltzmann constant and ${\dot{\varepsilon }}_{0t}$ is the reference strain rate (or frequency factor). Equation (2) shows the total activation energy related to the amorphous and crystalline phases altogether. To evaluate the activation energies of the crystalline and amorphous phases, individual reference strain rates should be considered, as follows:(3)$${\left.{\dot{\varepsilon }}_{c}={\dot{\varepsilon }}_{0c}\exp\left(\frac{-Q_{c}}{k_{B}T}\right)\right|}_{\sigma =const.}$$
(4)$${\left.{\dot{\varepsilon }}_{a}={\dot{\varepsilon }}_{0a}\exp\left(\frac{-Q_{a}}{k_{B}T}\right)\right|}_{\sigma =const.}$$

The reference strain rate of the crystalline phase based on the density of gliding mobile dislocations, *ρ*, controlled by pinning and the average dislocation glide velocity, *v_dis_*, can be obtained by [33,35]:(5)$${\dot{\varepsilon }}_{0c}=b\rho_{m}v_{dis}$$
where *b* is the Burgers vector of HDPE (=2.46 × 10^−10^ m), *ρ_m_* is the mobile dislocation density (=5 × 10^16^ m^−2^), and *v_dis_* is the velocity of dislocation (=1374 ms^−1^) [7]. With this information, ${\dot{\varepsilon }}_{0c}=1.7\times 10^{10}s^{-1}$. The reference strain rate of the amorphous phase is considered as ${\dot{\varepsilon }}_{0a}=1\times 10^{18}s^{-1}$ [36]. According to Lucas and Oliver [37], the strain rate can be approximated by the indentation depth ratio:(6)$$\dot{\varepsilon }=\frac{\dot{h}}{h}$$

Therefore, using the strain rate obtained by the nanoindentation tests and the reference strain rate values of the crystalline and amorphous phases, the activation energies as a function of time for crystalline and amorphous phases can be calculated.

Scatter in the creep rates was observed, which indicated that the activation energy exhibited oscillations aside from a certain average value. This is illustrated in Figure 5, showing the total activation energy, *Q*, versus time calculated using Equations (3) and (4). The activation energy of the crystalline phase obtained by Equations (3) and (5) is shown in the second, right peak of Figure 5. The scatter was relatively small and the mean value of the creep activation energy of the crystalline phase was 0.66 (±0.02) eV. To evaluate the creep activation energy of the amorphous phase, the reference strain rate of the amorphous phase for HDPE was used, which showed the mean value of the activation energy of the amorphous phase to be 0.38 eV. The obtained activation energy value for the generation of strain bursts in the crystalline phase by high-resolution nanoindentation testing was approximately twice the evaluated activation energy in the creep process of the amorphous phase. It should be noted that all values of the activation energy did have not equal probability and the distribution density of the activation barrier heights exhibited a noticeable maximum (Figure 5). This indicated that there were the most probable values of the activation energy. Hence, it was established that the distribution of the activation energies of creep in PE depended on the supramolecular structure of the polymer studied. This is illustrated in Figure 5, showing total activation energy, Q, versus time calculated using Equations (3) and (4). The activation energy of the crystalline phase obtained by Equations (3) and (5) is shown in Figure 5. The scatter of $\dot{\varepsilon }$ was relatively small and the mean value of creep activation energy of the crystalline phase for the first peak was 0.66 eV and 0.78 eV for the second peak. It should be noted that all values of the activation energy did have not equal probability and the distribution density of the activation barrier heights exhibited a noticeable maximum (Figure 5). This indicated that there were two maximum probable values of the activation energy. Hence, it was established that the distribution of the activation energies of creep in PE depended on the supramolecular structure of the polymer studied.

For another estimation of the activation energy of the crystalline phase, we used the Arrhenius equation, in which strain burst numbers at different temperatures are counted. The Arrhenius Equation (2) can also be written using the following expression:(7)$$\ln(k)=-\frac{Q}{k_{B}}\frac{1}{T}+\ln(A)$$
where *k* is the rate constant (number of net jumps per second) and *A* is the frequency factor (total number of fluctuations per second). In the creep rate curve, the numbers of net jumps exceeding a certain threshold, at different temperatures, were counted. Thus, according to the Arrhenius equation by plotting the number of net jumps versus reciprocal temperature, the activation energy can be obtained. It is important to note that a low thermal interval was considered to remain the constant coefficient of exponential Equation (2), in other words, the creep mechanism did not change. From the slope $(Q/k_{B})$ of the line, the activation energy Q can be obtained. The best fit for the results was calculated using the least-square method.

The experimental results indicated that the strain burst numbers observed during nanoindentation creep decreased at higher temperatures. This was due to the enhancement of the viscous flow of the amorphous phase in polyethylene. As shown in Figure 6, the ln(*k*) values (*k* is the number of net jumps per second) obeyed an Arrhenius law, indicating the mean value of the activation energy of the crystalline phase in high-resolution nanoindentation creep to be 0.64 (±0.01) eV for HDPE (the measured value Q is the heat of activation H for the dislocation movement). This result was in good agreement with the results from the previous method, using activation energy distributions by the Arrhenius equation, which was approximately 0.66 eV. Based on this value and the corresponding literature, a detailed discussion of the reaction mechanisms was possible.

### 3.2. Activation Volume

A study of the activation volume could provide important information about the dominant mechanisms of inelasticity in HDPE. The activation volume represents the volume of the polymer segment involved in polymer flow, i.e., the volume of the polymer segment that has to move as a whole to activate plastic deformation. The activation volume, v*, can be obtained by v* ≈
*l*bd [38], in which *l* is the obstacle’s distance, b is the Burgers vector magnitude (b = 0.246 nm for HDPE), and d is the activation distance (a fraction of the obstacle width), which is related to the size of the obstacle. In the case of repulsive forest-dislocation obstacles, it is on the order of the Burgers vector. Therefore, with further approximation, d ≈ b [39], so the activation volume becomes v* ≈
*l*b^2^ and the obstacle (pinning) distance, *l*, is considered the geometrically necessary dislocation (GND) mean free distance, which is given by [40,41]:(8)$$l=\frac{1}{\sqrt{\rho_{GND}}}$$

The Nix and Gao model has been used to measure geometrically dislocation density in HDPE by nanoindentation hardness experiments and was established using the Taylor dislocation model and a model of geometrically necessary dislocations underneath an indenter tip. Using this model, geometrically necessary dislocations can be estimated by the following equation [42]:(9)$${\bar{\rho }}_{GND}=\frac{1}{4bh}\tan^{2}\theta$$
where *h* is indentation depth, θ is the angle between the surface of the material and the surface of the indenter (20° in the present work for the Berkovich tip), and *b* is the Burgers vector of the dislocations (*b* = 0.246 nm for HDPE). Finally, using nanoindentation hardness tests, the geometrically necessary dislocations were estimated as *ρ_GND_* ≈ 0.5 × 10^16^ m^−2^. Substituting this value in Equation (8), *l* = 14.1 nm. Thus, the calculated value of the activation volume equals an activation volume of v* = 0.86 nm^3^ (≈58 b^3^ in atomic scale).

### 3.3. Local Strain Rate Sensitivity

The determination of strain rate sensitivity is an important material property in revealing and understanding thermally-activated plastic deformation mechanisms. It is often defined as the variation in hardness with the strain rate at a given strain and temperature, using the following equation [43,44]:(10)$${\left.m_{nanoindentation}=\frac{d(lnH)}{d(ln{\dot{\varepsilon }}_{indentation})}\right|}_{\varepsilon ,T}$$
where *m* is the strain rate sensitivity index, 0 < *m* < 1, which describes the strain rate sensitivity behavior of the material assuming a constant microstructure. A value of *m* = 0 describes a rigid-perfectly plastic material and *m* = 1 describes a linear viscous solid, respectively [45]. In Equation (10), it is assumed that the conditions approximate the steady-state process [46]. The value of strain rate sensitivity can be determined by using either strain rate jump tests during a single macroscopic test or experiments with several tests at different strain rates. Therefore, it can be obtained by mechanical testing and also be related to the activation volume. The following equation shows the relationship between strain rate sensitivity, m, and activation volume, *V**, as [24]:(11)$$m=\frac{2.64\sqrt{3}kT}{V^{*}\cdot H}$$
where *k* is the Boltzmann constant, *T* is the absolute temperature, *H* is the hardness, and *V** is the activation volume for the plastic deformation, which is directly related to the deformation mechanism. In this equation, the number 2.64 is related to the relationship between yield stress and the nano hardness value (*H* ≈ 2.64 *σ*_y_) [24]. HDPE samples were tested at different strain rates over a range of 6.2 × 10^−4^ s^−1^ and 6.3 × 10^−3^ s^−1^ (low strain rate). All tests were performed at 21.0 ± 0.1 °C. The ASMEC Universal Nanomechanical Tester UNAT was employed for the nanoindentation experiments on HDPE samples using a Berkovich indenter with a 300 nm Berkovich tip with loads of 4–5 mN with a step of 0.1 mN to obtain a range of strain rates. Unloading rates were set equal to loading rates. In all tests, there was an additional hold of 60 s at 90% unloading to assess creep recovery. The strain rate sensitivity was determined by nanoindentation creep experiments at which HDPE creeps. The constant displacement-rate ($\dot{h}$) method was used, in which the histories of hardness and strain rate were monitored. Thus, the m value could be estimated from the slope of the linear fit of logarithmic hardness versus linear strain rate in Figure 7. The slope of this linear curve, as shown in Equation (10), was the strain rate sensitivity (m). By obtaining (m) from Equation (10) and inserting it into Equation (11), the amount of activation volume (*V**) was obtained. Thus, the activation volume obtained by this method was *V**= 0.92 (nm)^3^ (~60 b^3^), which was in good agreement with the result obtained using the approximation method in the previous section. In order to obtain an idea of the scale of this volume, it is important to note that the unit cell volume of HDPE contains 4 methylene units and the volume of the polyethylene unit cell with the space group notation Pnam-D_2h_ is 0.495 nm × 0.253 nm × 0.740 nm = 0.093 nm^3^ [47]. Therefore, the activation volume for the HDPE is a volume equivalent to ≈40 CH_2_ units. Due to the small activation volume of the creep process in HDPE, it can be concluded that a local molecular process caused the breakage of the intercrystalline tie molecule entanglement.

## 4. Discussion

There are many papers addressing the creep behavior of polyethylene. However, the reported activation energies vary greatly from study to study. Sinclair and Edgemond [48] found an activation energy of 0.53 eV (=50.8 kJ/mol) with small stress extrapolated to zero stress by studying the creep of polyethylene using the conventional method. Thornton [49] performed creep tests on polyethylene at elevated temperatures and obtained an activation energy of 1.2 eV (=115.8 kJ/mol). Chaney [50] and Govaert [51] used time-temperature superposition to evaluate oriented HDPE materials, reporting activation energy values of 1.04 eV (=100 kJ/mol) and 1.19 eV (=115 kJ/mol), respectively. Zhou et. al. [52] evaluated the activation volume for HDPE films on the order of 1–2 nm^3^, reporting activation energies within a span of 0.21–0.31 eV and believing that this was connected to deformation of the amorphous phase, more specifically, to the tie-chains.

Li [1] found an activation energy of 0.22 eV for the generation of strain bursts in the crystalline phase by studying the nanoindentation creep of HDPE at temperatures ranging from 30 to 70 °C, obtaining an activation volume of 0.22 nm^3^ at a temperature of 30 °C. However, because the low activation energy of 0.22 eV was even lower than the activation energy of the amorphous phase at 0.38 eV, the result can not be reasonable. Wilhelm in 2017 [25] carried out high-resolution torsion creep tests using a rheometer on bulk samples of HDPE. His investigations exhibited marked strain bursts, very similar to the strain bursts observed by nanoindentation creep tests in this work, and initially obtained the activation energy of the crystalline phase to be approximately 0.59 eV, and finally reported a value of 0.65 eV. The activation energy of the amorphous phase was measured as being 0.31 eV, while the activation volume of the amorphous phase was found to be 1 nm^3^. All of these results are in good agreement with our work. Argon [53] analyzed the nucleation of dislocation half-loops from lamellar faces in polyethylene and concluded that for a wide range of stresses, the activation energy for half-loop generation was on the order of 1 Gb^3^ (where G is shear modulus) and was equal to 0.052 eV for HDPE. This value was significantly smaller than the present activation energy (0.64 eV). Argon also calculated the activation volume on the order of 10^2^ b^3^ to be 1.5 nm^3^, which was 60% larger than the values reported in Wilhelm’s study and the current experiments. Peterson [54] suggested that screw dislocations are generated from the edges of the lamellae under shear stress and/or due to thermal fluctuations, and that the thermal energy fluctuation necessary to generate a dislocation of 20 nm length was between 0.5 eV and 1 eV. The path length in semicrystalline polymer is known to be limited by the lamellar thickness, which in our case was approximately 20 nm, and all strain bursts were less than this value. Therefore, according to the obtained results (activation energy of 0.64 eV), Peterson’s model was more acceptable within the frame of the results of this study. In the following, Table 1, the activation energies and activation volumes of the crystalline and amorphous phases of HDPE (Q_c_, V_c_, Q_a_, V_a_) obtained in this study are compared with the results obtained by Li et al. and Wilhelm.

Here is a further interpretation provided by a comparison with the standard relaxation processes established in the literature. Polyethylene exhibits three mechanical relaxations, designated as the α, β, and γ processes, each of which occurs within a certain range of temperature [55]. The γ process is due to the localized motions of either chain ends or branches associated with the amorphous phase [56,57], although originally it was also proposed to arise from the crystalline phase. It is usually observed in the temperature range of −150 to −120°C (123–153 K) [58,59]. The β process, commonly occurring in the amorphous phase of PE, occurs in the temperature range of −30 to +10 °C (243–283 K). Takayanagi et al. [60,61] generally proposed that the deformation of bulk crystallized polyethylene is composed of: (1) the α_I_ process, represented by the deformation in the intermosaic segment related to the conformational change of distorted molecular chains of this segment; (2) the α_II_ process, consisting of uniform c-axis shear deformation of lamellar crystals; and even by (3) the β process, consisting of interlamellar slip, i.e., deformation of interlamellar amorphous section. Each of these processes predominates according to the temperature range mentioned above. These relaxation mechanisms (α_I_, α_II_, and β processes, respectively) are shown in Figure 8.

Most relevant to the experiments of this study is the mechanical αprocess because it occurs within the temperature range of +30 to +120 °C (303–393 K) [55,56]. This process is observed in all semicrystalline polymers [57] and it is commonly agreed that it arises from the crystalline phase and—more specifically—from an intralamellar process, i.e., motions of chain units therein within the temperature range. This is also highly relevant for most of the experiments of this study. Different molecular mechanisms have been suggested to interpret this relaxation process, rather than being caused by the distribution of crystalline lamellar thickness, such as rotation of the crystalline sequences followed by a translation along the chain axis, or torsional twisting in the crystalline sequence, among others [59,62]. Two α processes, called α_I_ and α_II_, have even been identified [63,64]. The α_I_ process is regarded as an intralamellar process that is related to a slip mechanism along a block grain boundary and/or the reorientation of crystal grains within the crystalline lamellae around the a- and b-axes [65]. The β—and γ— processes cannot apply to the current case of polyethylene nanoindentation creep performed at room temperature, as they occur at temperatures far below room temperature. Concerning α relaxation, only the α_I_ mechanism takes place around room temperature, while the α_II_ mechanism only occurs at temperatures higher than room temperature. The activation energy for α_II_ relaxation is higher than that for α_I_ relaxation, ranging from 1.5 to 2 eV. The calculated activation energies for both mechanisms become lower with increasing deformation [66,67]. In general, the values of activation energy for the α process given in the literature vary largely from study to study, ranging from 0.79 to 1.1 eV [1,56,64,68]. Laredo et.al. [56] found an activation energy of HDPE of 0.9 eV by using the dynamic mechanical method as well as the thermally stimulated depolarization technique. Zubova et.al. [57] obtained an activation energy for PE crystals to be 0.82 eV using molecular dynamics (MD) simulation. Suljovrujic et.al. [69] evaluated the activation energy for the HDPE to be on the order of 1.1 to 0.79 eV using dielectric spectroscopy before and after deformation. In particular, the latter values are close to the activation energy of deformed HDPE evaluated within this study, which showed that the mean value of the activation energy of the crystalline phase was 0.66 eV. Therefore, the activation process found in this work to act in the crystalline phase can be interpreted as the α_I_ process. This process is described in the literature as an intracrystalline process that occurs when the block grain boundary starts to glide within the crystalline lamellae by longitudinal chain transport around the a- and b-axes. It may be connected with the movement of a screw dislocation, even in combination with a repeated dislocation stop/tear-off process and the occurrence of dislocation avalanches in the wake of a block grain boundary.

## 5. Conclusions

Investigations of the deformation behavior of HDPE on a small scale by measuring quantities such as the activation energy and activation volume offer a way to improve the understanding of the large-scale mechanical and physical properties of crystalline polyethylene. The nanoindentation creep tests on HDPE showed that, in general, the macroscopic plastic deformation appeared as a smooth and uniform curve, but that at certain loads and loading rates, discrete strain jumps and/or jerky motions in the creep curve occurred. The significant strain jumps were interpreted in terms of breaking avalanches of dislocations followed by the spreading of dislocations and/or dislocation groups occurring in the crystalline phase due to the following reasons:In previous work [24], nanoindentation creep studies applied to as-received HDPE samples revealed the occurrence of strain bursts. The observations were interpreted in terms of breaking avalanches of dislocations. Analysis of the experimental data revealed that the number of jumps and the jump size not only depended strongly on the load but also on the loading rate. These results indicated that the occurrence of strain bursts in nanoindentation creep experiments of polyethylene was related to crystalline relaxation events.The present experimental results indicated that a thermally activated dislocation-mediated plasticity mechanism was active in HDPE at room temperature. The presently observed activation energy of 0.66 eV and activation nucleus with a length of 20 nm best fit Peterson’s model, predicting 0.5–1.0 eV for the nucleation and motion of a dislocation segment with a length of 20 nm.

Therefore, it can be concluded that the involvement of dislocation avalanches in plastic deformation is highly probable and the occurrence of strain bursts in nanoindentation creep experiments of HDPE is related to crystalline relaxation events.

A further interpretation can be added here in terms of the α_I_ relaxation mechanism, which also occurs at room temperature: Its activation energy was close to the crystalline activation energy calculated for the nucleation/mobilization of dislocations (0.66 eV), especially bearing in mind that the activation energy for the α_I_ mechanism decreases with increasing deformation. The α_I_ process in the literature is described as an intracrystalline process where the block grain boundaries within a lamella start to slide via longitudinal chain transport, corresponding to the creation and motion of a screw dislocation around the a- and b- crystallographic axes. This means that the dislocation nucleation and motion mechanism identified according to Peterson’s model may be due to the creation of the screw dislocation mentioned above and with its motion along a block grain boundary.

To conclude, high-resolution nanoindentation creep experiments on polyethylene proved to be a successful method for the investigation of crystalline phase deformation mechanisms via observation and quantitative analysis of strain bursts. Although such studies are well known with metals, this phenomenon has only recently been investigated in semicrystalline polymers using nanoindentation creep testing, and only in HDPE to date. There is still much work to be done in support of the dislocation approach, both experimentally and theoretically.

## Figures and Tables

**Figure 1 materials-16-00143-f001:**
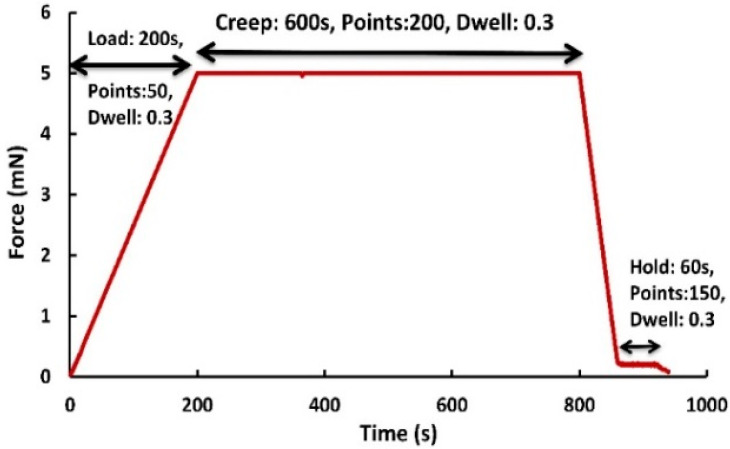
Four-step scheme of the load-hold-unload-hold nanoindentation test showing force-time characteristics.

**Figure 2 materials-16-00143-f002:**
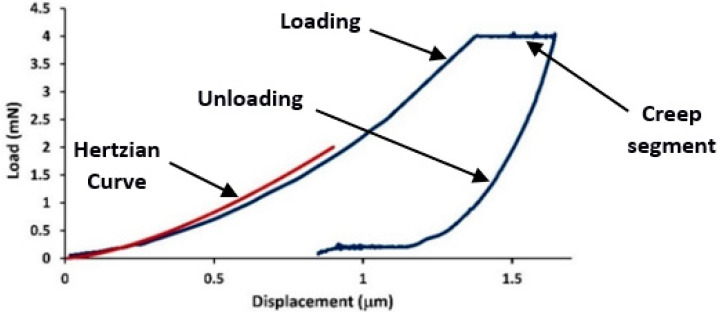
Representative nanoindentation curve, indicating indentation load in terms of indentation depth obtained by the load–hold–unload–hold nanoindentation test on HDPE. Elastic response according to Hertzian elastic contact theory (red line) [32] and experimental nanoindentation curve (blue line) for a load of 4 mN and loading rate of 25 µN/s.

**Figure 3 materials-16-00143-f003:**
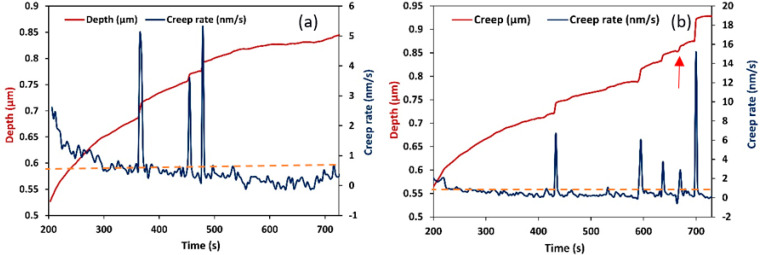
Two representative creep curves of HDPE, (**a**,**b**), showing penetration depth (red line) and creep rate (blue line) as a function of time during the creep segment through 500 s.

**Figure 4 materials-16-00143-f004:**
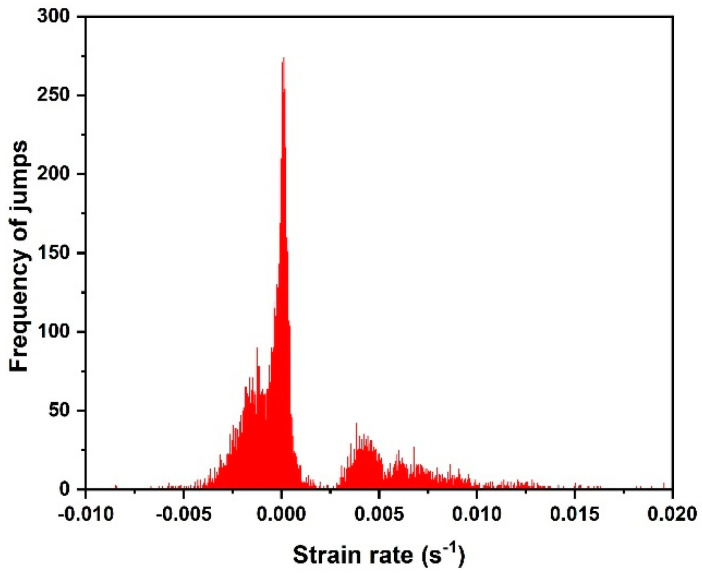
The number/frequency distribution of jumps within both crystalline and amorphous phases as a function of the strain rate.

**Figure 5 materials-16-00143-f005:**
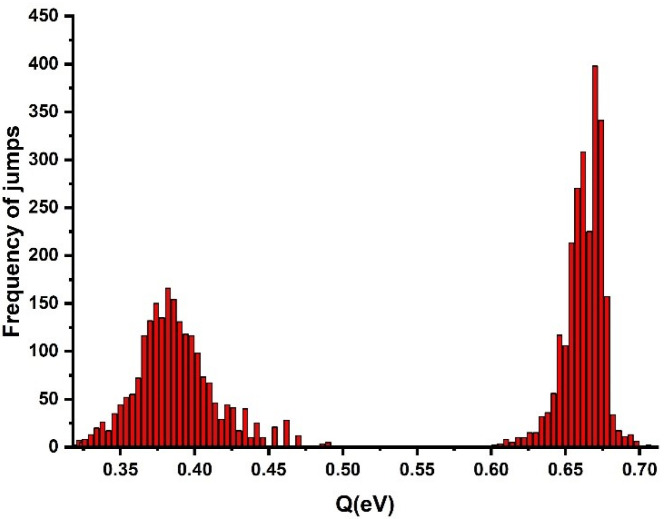
The amorphous and crystalline activation energy distributions, in which the first peak shows the mean value of the activation energy of the amorphous phase (0.38 eV) and the second peak shows the mean value of the activation energy of the crystalline phase (0.66 eV).

**Figure 6 materials-16-00143-f006:**
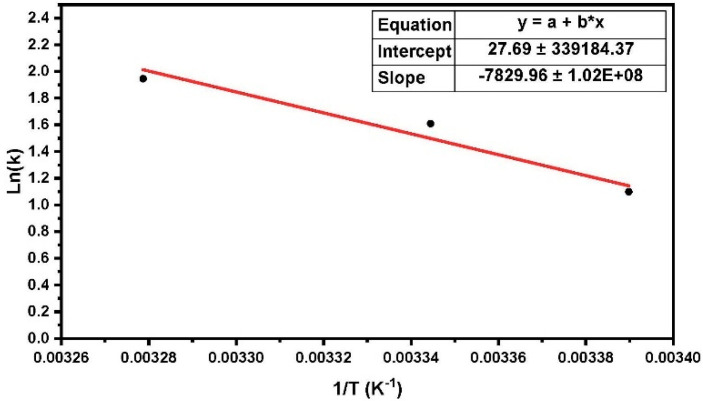
Arrhenius plot of the strain burst in HDPE with 45.3% crystallinity. The value of the activation energy obtained at different temperatures was 0.64 eV (in the figure, multiplication denote by an asterisk *).

**Figure 7 materials-16-00143-f007:**
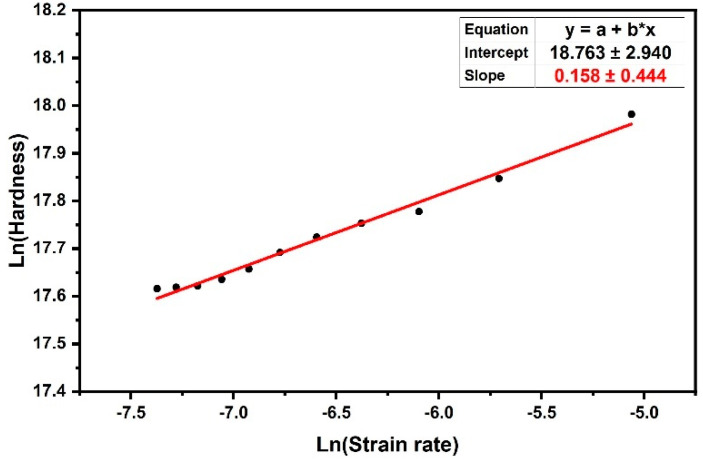
The logarithm of hardness plotted versus the logarithm of strain rate for 51 creep nanoindentation tests on HDPE, for which the value of strain-rate sensitivity was *m* = 0.158 ± 0.001. The value of the activation volume obtained by this plot was 0.92 (nm)^3^.

**Figure 8 materials-16-00143-f008:**
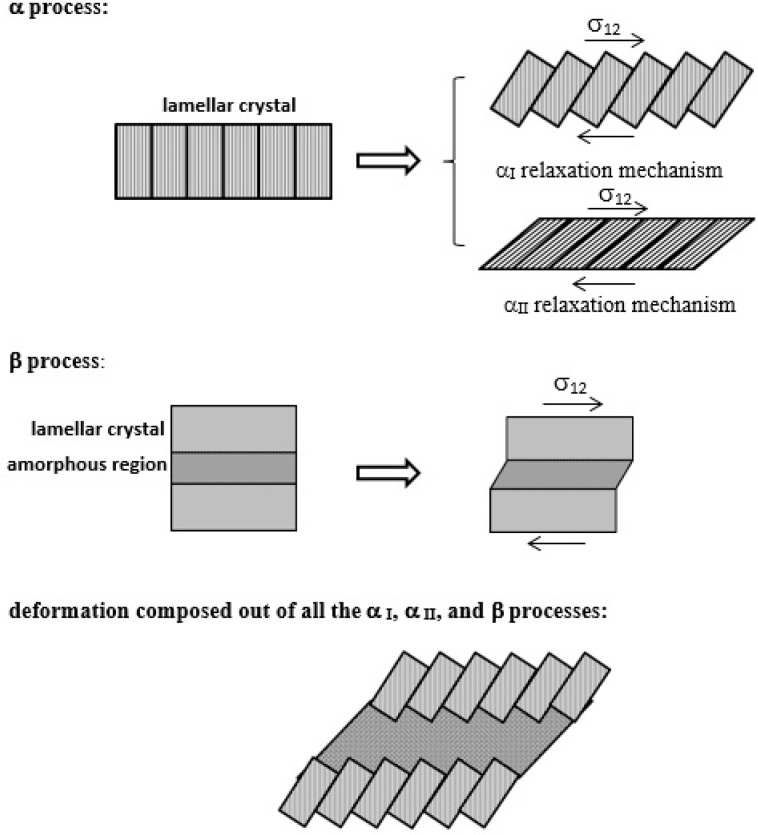
Schematic representation of three types of deformation in polyethylene. From top to bottom: the intermosaic block deformation (α_I_), the interlamellar c-axis shear deformation (α_II_), the interlamellar deformation (β), and the deformation composed of all α_I_, α_II_, and β processes [61].

**Table 1 materials-16-00143-t001:** Comparison of the activation energies and activation volumes of the crystalline and amorphous phase of HDPE (Q_c_, V_c_, Q_a_, V_a_) obtained in this study with the results obtained by Li et al. and Wilhelm.

	This Work		Li, J. and Ngan, A. (2010)	Wilhelm, H. (2017)
Q_c_ (eV)	0.66 Crystalline activation energy distributions	0.64 Arrhenius plot of strain bursts induced by nanoindentation creep test	0.22 Arrhenius plot of strain bursts induced by nanoindentation creep test	0.59 & 0.65 Arrhenius plot of strain bursts induced by torsion creep test
V_c_ (nm^3^)	0.8 Approximation of V* = lb^2^		0.22 Fitting data to Arrhenius eq.	-
Q_a_ (eV)	0.38 Amorphous activation energy distributions		-	0.31 Evaluation of average creep rate
V_a_ (nm^3^)	0.92 Local strain rate sensitivity		-	1 By average creep rate

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
