# Peer review of "The Activation Energy of Strain Bursts during Nanoindentation Creep on Polyethylene"

_materials, 2022, doi:10.3390/ma16010143_

Round 1
Reviewer 1 Report
The paper presents the activation energy of strain bursts during nanoindentation creep on polyethylene. The article is scientifically interesting.
Please:
1. verify the article linguistically
2. provide literature references (in the case of the mathematical formulas taken from the literature)
3. standardize the description of the axis / units in the Figures
4. describe and present the repeatability / errors of experiments
Reviewer 2 Report
The authors have addressed an interesting manuscript about the determination of the activation energy of strain bursts during creep in HDPE. This investigation is a very conceptual study, which will be interesting by readers. However, due to the quality of the manuscript, I am affraid that major revisions are suggested.
The main problem is that Greek characters appears as @, which is not a big issue until we start reading the Discussion. It is not possible to follow at all and it is not understandable. I am looking forward to the revised version to see if the conclusions are supported by their results.
Also, I am concerned about the validity of this methodology. I would like the authors to explain in depth the temperature issue. The range presented in Figure 6 (21-32 ºC) is not of huge interest to the scientific community.
There are also some minor comments:
- "The experimental results indicated that the strain burst numbers observed during nanoidentation creep decreases at high temperatures". Using "high temperatures" in this sentence is not appropiate since the tests are conducted at 32 ºC. HDPE in greenhouse applications reaches temperatures higher than 32 ºC in the Mediterranean, Australian or Californian areas. I suggest to change this therminology.
-Equation 6 has been developed for metals. I suggest to the authors to comment that this is an approximation or give further information about the validity in thermoplastics.
Round 2
Reviewer 2 Report
The authors have addressed all changes and the manuscript is suitable for publication in Materials.